# Q-Cell Glioblastoma Resource: Proteomics Analysis Reveals Unique Cell-States Are Maintained in 3D Culture

**DOI:** 10.3390/cells9020267

**Published:** 2020-01-21

**Authors:** Rochelle C. J. D’Souza, Carolin Offenhäuser, Jasmin Straube, Ulrich Baumgartner, Anja Kordowski, Yuchen Li, Brett W. Stringer, Hamish Alexander, Zarnie Lwin, Po-Ling Inglis, Rosalind L. Jeffree, Terrance G. Johns, Andrew W. Boyd, Bryan W. Day

**Affiliations:** 1Cell and Molecular Biology Department, QIMR Berghofer Medical Research Institute, Sid Faithfull Brain Cancer Laboratory, Brisbane, QLD 4006, AustraliaCarolin.Offenhauser@qimrberghofer.edu.au (C.O.); Ulrich.Baumgartner@qimrberghofer.edu.au (U.B.); Anja.Kordowski@qimrberghofer.edu.au (A.K.); Michelle.Li@qimrberghofer.edu.au (Y.L.); Brett.W.Stringer@gmail.com (B.W.S.); Hamish.Alexander@health.qld.gov.au (H.A.); Zarnie.Lwin@health.qld.gov.au (Z.L.); Po-Ling.Inglis@health.qld.gov.au (P.-L.I.); Lindy.Jeffree@health.qld.gov.au (R.L.J.); Andrew.Boyd@qimrberghofer.edu.au (A.W.B.); 2Immunology Department, QIMR Berghofer Medical Research Institute, Gordon and Jessie Gilmour Leukaemia Research Laboratory, Brisbane, QLD 4006, Australia; Jasmin.Straube@qimrberghofer.edu.au; 3Royal Brisbane and Women’s Hospital, Brisbane, QLD 4006, Australia; 4Telethon Kids Institute, Perth, WA 6009, Australia; Terrance.Johns@telethonkids.org.au; 5School of Biomedical Sciences, The University of Queensland, Brisbane 4072, Australia; 6School of Biomedical Sciences, Faculty of Health, Queensland University of Technology, Brisbane 4059, Australia

**Keywords:** glioblastoma (GBM), proteomics, GBM cell-states, recurrence, tumour heterogeneity, therapeutic targets

## Abstract

Glioblastoma (GBM) is a treatment-refractory central nervous system (CNS) tumour, and better therapies to treat this aggressive disease are urgently needed. Primary GBM models that represent the true disease state are essential to better understand disease biology and for accurate preclinical therapy assessment. We have previously presented a comprehensive transcriptome characterisation of a panel (n = 12) of primary GBM models (Q-Cell). We have now generated a systematic, quantitative, and deep proteome abundance atlas of the Q-Cell models grown in 3D culture, representing 6167 human proteins. A recent study has highlighted the degree of functional heterogeneity that coexists within individual GBM tumours, describing four cellular states (MES-like, NPC-like, OPC-like and AC-like). We performed comparative proteomic analysis, confirming a good representation of each of the four cell-states across the 13 models examined. Kyoto Encyclopedia of Genes and Genomes (KEGG) pathway analysis identified upregulation of a number of GBM-associated cancer pathway proteins. Bioinformatics analysis, using the OncoKB database, identified a number of functional actionable targets that were either uniquely or ubiquitously expressed across the panel. This study provides an in-depth proteomic analysis of the GBM Q-Cell resource, which should prove a valuable functional dataset for future biological and preclinical investigations.

## 1. Introduction

Glioblastoma (GBM) is the most common and aggressive form of adult brain cancer, and patients have not experienced significant increases in overall survival for several decades [1,2,3]. This lack of progress is, in part, attributable to the significant heterogeneity that exists within these tumours [4,5,6,7,8]. Recent seminal studies by Suva and colleagues have discovered that multiple dynamic cell-states (MES-like, NPC-like, OPC-like and AC-like) can reside within a single tumour entity, highlighting the degree of heterogeneity and plasticity present within GBM [9]. Therefore, therapeutic strategies with the ability to target each functional state have the potential to produce more durable clinical responses. Another significant factor preventing successful disease translation has been the lack of faithful models to drive meaningful neuro-oncology discoveries. To address this problem, we embarked upon the generation and characterisation of a panel of clinically relevant primary in vitro and in vivo GBM models (termed Q-Cell) [10,11]. Our previous Q-Cell characterisation studies have primarily focused on transcriptome expression analysis using microarray and RNA Sequencing (RNA-Seq) approaches. This study sought to expand upon our previous transcriptome findings by better understanding the proteomic profile of these models. Relative protein abundances of 6167 human proteins were quantitated using mass spectrometry (MS) and their expression patterns analysed bioinformatically. Principal component analysis, conducted on independent Q-Cell replicates, revealed consistent protein expression within a given model. These data indicated a high degree of fidelity at the protein level. All Q-Cell GBM models have been generated from IDH1 WT (wild-type) disease specimens. Not surprisingly, given this IDH1 wild-type (WT) status, we observed a high degree of correlation when comparing total protein expression between each model. Correlation at the protein level was most pronounced in a pair-matched pre- and post-treatment GBM specimen (SB2/SB2b). We also sought to analyse the relevance of the recently described GBM cell-states in our Q-Cell resource. Proteomics analysis revealed a good representation of each of the four cell-states across the 13 models examined. We also characterised a number of hybrids (n = 5) that displayed significant heterogeneity with elevated protein expression of multiple cell-state markers within a given model. This finding was consistent with the Suva study, where the single-cell RNA-Seq data showed a significant number of tumour cells in transition states [9]. Despite a small sample size, detailed analysis of the single pair-matched model SB2/SB2b highlighted a transition at recurrence to a more MES-like state. This is consistent with the findings of others, indicating an increase in mesenchymal disease when GBM tumours recur post-therapy [4,12]. Furthermore, of the 257 cell-state-characterisation genes assessed, we detected 153 (57%) at the protein level. This finding indicated reasonably high levels of mRNA translation, providing significant scope for future biological and preclinical therapy studies aimed at targeting multiple GBM cell-states.

Unsupervised hierarchical clustering identified that eight of the models tended to express structural proteins associated with a broad mesenchymal-like signature, while the remaining five models expressed proteins associated with a more neuronal-like GBM signature. Kyoto Encyclopedia of Genes and Genomes (KEGG) pathway analysis identified upregulation of a number of GBM-associated cancer pathway proteins across the 13 models, as expected. We also undertook bioinformatics analysis, using the OncoKB database, to identify functional actionable targets that were either uniquely or ubiquitously expressed across the models.

The present effort has sought to provide a detailed proteomics analysis and build upon our previously described transcriptomic analysis of the Q-Cell GBM resource. Findings of this study have increased our understanding of GBM tumour heterogeneity and cell-states at the protein level and further validated our previous mRNA datasets. Given the highly functional nature of proteomics analysis, we hope this freely available GBM resource and combined datasets will provide a highly relevant functional platform to drive meaningful biological and translational neuro-oncology discoveries into the future.

## 2. Materials and Methods

### 2.1. Cell Culture

We have developed a characterised GBM patient-derived cell line resource (Q-Cell) [10,11,12,13], in which lines are maintained as glioma neural stem cell (GNS) cultures [14] or as 3D neurosphere cultures [15] using StemPro NSC SFM (Invitrogen, Carlsbad, CA, USA) or KnockOut™ DMEM (Gibco, Thermo Fisher Scientific, Waltham, MA, USA) as per the manufacturer’s guidelines. Characterisation data is freely available from https://www.qimrberghofer.edu.au/q-cell/. All tissues were collected following ethical approval from the Royal Brisbane Women’s Hospital and QIMR Berghofer Human Research Ethics Committees. Ethical approval number: P3420, HREC/17/QRBW/577 Novel Therapies for Brain Cancer. In order to maintain pluripotency, KnockOut™ DMEM (Gibco) media were supplemented with GlutaMAX™ Supplement (Gibco), StemPro™ Neural Supplement (Gibco), Recombinant Human EGF (Gibco), Recombinant Human FGFb (Gibco), and Penicillin/Streptomycin (Gibco). Cells were cultured on flasks coated with Basement Membrane Matrigel® Matrix (Corning, New York, NY, USA). Passaging the cells was done by detaching the cells from the flask surface using Accutase® solution (Sigma-Aldrich, St. Louis, MO, USA). Glioma cells were cultured as glioma neural stem (GNS) cultures as outlined in detail in [14] or as tumourspheres using StemPro^®^ NSC SFM. 

### 2.2. Proteome Sample Preparation

GBM cells (5 × 10^6^) were seeded in T75 flasks with StemPro^®^ NSC SFM (Thermo Fisher Scientific, Waltham, MA, USA). Cell lysates were collected in biological triplicates 72 hours postseeding. Cells were lysed in buffer containing 2% SDS (Sigma-Aldrich), EDTA-free protease inhibitor (Roche, Basel, Switzerland) and 100 mM tris-HCl (pH 7.6) [16]. The lysates were sonicated and incubated for 5 min at 95 °C, protein concentration was quantified, and samples were processed by an in-solution digest protocol. Briefly, 100 µg of cell lysate was precipitated overnight with 5 volumes of ice-cold acetone. The precipitate was collected by centrifugation and resuspended in 8 M urea (Sigma-Aldrich). The samples were reduced with dithiothreitol (Sigma-Aldrich) for 1 h at room temperature, alkylated with iodoacetamide (Sigma-Aldrich) in the dark, diluted to a final concentration of 2 M urea, 10% acetonitrile (Merck, Kenilworth, NJ, USA) and 1 µg of trypsin (Promega, Madison, WI, USA) and digested to peptides overnight at 37 °C. The peptides were desalted on C18 stage tips as described [17] and resuspended in 0.1% formic acid.

### 2.3. Liquid Chromatography (LC) and Mass Spectrometry (MS) Analysis

Peptides were injected under trapping conditions using a Waters (Milford, MA, USA) NanoAcquity system interfaced to a linear triple quadrupole (LTQ)-Orbitrap elite mass spectrometer (Thermo Fisher Scientific). Acidified digested samples were loaded onto a Waters 2G-V/M C18 Symmetry trap (5 µm particle size, 180 µm × 20 mm) at 5 µL/min in 99% solvent A (0.1% (v/v) aqueous formic acid) and 1% solvent B (100% (v/v) acetonitrile containing 0.1% (v/v) formic acid) for 3 minutes, and the peptides were subsequently separated inline using a pre-equilibrated analytical column (Waters C18 BEH 130 Å, 1.7 µm particle size, 75 µm × 200 µm) at a flow rate of 0.3 µL/min and temperature of 35 °C. The gradient ran at 300 nL/min from 2% B (0.1% formic acid in acetonitrile) to 40% B over 190 min, rose to 95% B to wash the column, then re-equilibrated at 2% B for the next injection. Eluate was delivered into the mass spectrometer via a Nanospray Flex Ion Source (Thermo Fisher Scientific) containing a 10 μm P200P coated silica emitter (New Objective, Woburn, MA, USA). Typical spray voltage was 1.8 kV with no sheath, sweep, or auxiliary gases; the heated capillary temperature was set to 285 °C. The LTQ-Orbitrap Elite mass spectrometer (Thermo Fisher Scientific) was controlled using Xcalibur 2.2 SP1.48 software (Thermo Fisher Scientific) and operated in a data-dependent acquisition mode to automatically switch between Orbitrap-MS and collision induced dissociation (CID)- based ion trap-MS/MS. The survey full scan mass spectra (from m/z 380 to 1700) were acquired in the Orbitrap with a resolving power of 120,000 after accumulating ions to an automatic gain control (AGC) target value of 1.0 × 10^6^ charges in the LTQ. MS/MS spectra were concurrently acquired in the LTQ on the 20 most intense ions from the survey scan, using an AGC target value of 1.0 × 10^4^. Charge state filtering ( unassigned precursors and singly charged ions were not selected for fragmentation) and dynamic exclusion (repeat count 1, repeat duration 30 s, exclusion list size 500, exclusion duration 90 s) were used. CID fragmentation conditions in the LTQ were 35% normalized collision energy, activation q of 0.25, 10 ms activation time, and minimum ion selection intensity of 5000 counts. Maximum ion injection times were 200 ms and 50 ms for survey full scans and MS/MS scans, respectively.

### 2.4. Data Processing and Analysis

Spectra were analyzed using MaxQuant [18] version 1.5.3.30, and Andromeda [19]. The MS2 spectra were searched against the UniProt FASTA database version 31/01/2013. Enzyme specificity was set to trypsin, allowing for cleavage N-terminal to proline and between aspartic acid and proline. The search included carbamidomethylation of cysteine as a fixed modification and N-acetylation and oxidation of methionine as variable modifications. Up to two missed cleavages were allowed for protease digestion, and enzyme specificity was set to trypsin, defined as C-terminal to arginine and lysine excluding proline. The *identify* module in MaxQuant was used to filter (1% false identification rate (FDR)) identifications at the peptide and protein level. The identity of precursor peptides present in MS1, but not selected for fragmentation and identification by MS2 in a given run, was obtained by transferring peptide identifications based on accurate mass and retention times across liquid chromatography–mass spectrometry (LC–MS) runs where possible using MaxQuant [20]. Protein identifications were collapsed to the minimal number that contained the set of identified peptides. Proteome quantification was performed in MaxQuant using the extracted ion chromatography (XIC)-based label-free quantification (LFQ) algorithm [21]. In MaxQuant, a quantification event was reported only when isotope pattern could be detected and was consistent in terms of charge state of peptide. For quantification, intensities were determined as the intensity maximum over the retention time profile. Intensities of different isotopic peaks in an isotope pattern were summed up for further analysis. All RAW files and protein-based quantification results are available for download from the Q-Cell website at https://www.qimrberghofer.edu.au/q-cell/. 

### 2.5. Bioinformatics and Statistical Analysis

Bioinformatics analyses were performed using Perseus in MaxQuant [22]. Proteins identified on the basis of at least one unique peptide were used for all subsequent analyses. We selected the normalized abundances of proteins that were quantified in duplicates from at least one cell line. For comparing differences between all cell lines, biological triplicates were grouped by cell line, and the analysis of variance (ANOVA) was performed. We used the ANOVA method with largest power, permutation-based FDR of 0.05, and at least 250 repetitions for truncation. A two-sided student’s t-test was used to perform the comparison between two cell lines SB2 and SB2b employing a *p*-value cutoff of 0.05.

Unsupervised hierarchical clustering was performed on the z-score-transformed protein intensities or the z-scored log2 normalised mRNA read counts. Proteins were annotated with GO (Gene Ontology) biological process terms [23] and KEGG (Kyoto Encyclopedia of Genes and Genomes) pathway terms [24]. Statistical analysis of the enrichment of proteins with functional annotations was evaluated using a Fisher’s exact test (*p*-value <0.05). An enrichment score of greater than 1 indicates enrichment, and a score from 0 to 1 indicates negative enrichment. For comparison of mRNA and protein expression of the GBM-associated genes, the normalized protein intensities and normalized mRNA read counts were matched by gene name, and Pearson’s coefficient of correlation was calculated.

### 2.6. Cell-State ssGSEA

NPC1, NPC2, MES1, MES2, APC, and OPC marker genes were extracted from [9]. NPC1 and NPC2 were combined into a single gene set called NP, and MES1 and MES2 were combined into MES. z-score-scaled protein intensities of the models were assessed for NPC, MES, APC, and OPC enrichment using the R GSVA library (v1.32) with the ssGSEA algorithm as developed by Barbie et al. (2009) [25]. Principal component analysis was performed on scaled and centered ssGSEA enrichment scores. Replicates were summarised using an ellipse with the Euclidean distance from the centre and radius of 1. Heat maps were generated on the average ssGSEA enrichment scores of the cell line replicates. Hierarchical clustering of models and marker gene sets was performed on the 1-Spearman correlation dissimilarity matrix using the ward.D2 algorithm. A similar analysis was performed on the log2-normalised mRNA read counts from the previous transcriptomics study.

### 2.7. Cytoscape-Based Data Visualisation

Proteins identified from the list of metamodule genes describing the six different cell-states (MES1-, MES2-, NPC1-, NPC2-, OPC-, and AC-like) were visualised as nodes and grouped by the respective cell-states. z-score-scaled protein intensity values were plotted using continuous colour mapping of the nodes in Cytoscape version 3.7.2 [26]. For enrichment analysis, the top 100 significantly regulated proteins with the highest fold change absolute value were analysed for enrichment of gene ontology (GO) molecular functions, GO biological processes, and Kyoto Encyclopedia of Genes and Genomes (KEGG) pathways using the ClueGO App version 2.5.5 [27] in Cytoscape version 3.7.2 [26]. Analysis parameters were set to include GO terms (level 3 to 8), and KEGG pathways for which at least 5 proteins and 4% of associated proteins per term/pathway were present in the list of regulated proteins. A two-sided hypergeometric test with Benjamini–Hochberg correction was performed to identify significantly enriched/depleted terms (*p*-value <0.05) against a custom reference set consisting of all proteins identified in the SB2 and SB2b MS datasets. The kappa score threshold for clustering of related terms was set to 0.4.

## 3. Results

### 3.1. Q-Cell Proteomics Analysis

To accurately quantify proteome expression in the Q-Cell GBM panel, we performed label-free quantitative MS analysis of protein lysates from GBM cells grown in 3D neurosphere suspension cultures [15]. Primary models were initially grown as adherent GNS cultures on matrigel under serum-free conditions [14]. In order to avoid confounding our results with high abundance proteins present in the basement membrane of matrigel, cells were grown in 3D suspension cultures as neurospheres for 72 hours before lysis. We have previously shown that GBM cells either grown as neurospheres or on a substrate of matrigel tend to display equivalent characteristics [28]. Cell lysates were subjected to a defined proteomics workflow (Figure 1A). Peptides were analysed with 4 hour gradients on an LTQ-Orbitrap Elite mass spectrometer with CID fragmentation. Combined analysis of the spectra from all samples resulted in the identification of 6180 proteins at FDR of 1%. Using the label-free quantification algorithm in MaxQuant [21], we measured the abundance of 6167 proteins in at least one pairwise comparison (Appendix A). Analysis of triplicate samples showed excellent reproducibility when compared using principal component analysis of all quantified proteins (Figure 1B). Next, we compared the expression of the proteome by calculating the Pearson correlation coefficient (*r*) between each model. As expected, the highest correlation (r = 0.97) was observed between the pair-matched primary and recurrent model (SB2 and SB2b), while the least similar model identified was PB1 (r = 0.82) (Figure 1C).

We next measured the number of proteins identified in each of the triplicate runs from each Q-Cell model. Proteomics analysis identified an average of 2200 common proteins between each model. PB1 and SJH1 expressed the largest number of unique proteins (145 and 85 respectively) and therefore correlated the least to the remaining 11 models (Figure 1D and Appendix A). Some of the uniquely expressed genes include *CD44* (quantified in FPW1), *ERBB2* (quantified in RN1), and *OLIG2* (quantified in PB1) and are genes with key roles in brain cancer. *CDKN1A* and *CDKN2A* were highly expressed in RKI1 and detected in JK2 and MMK1; these cell lines are the only ones in our panel which do not have deletion of the coding gene at the genome level.

### 3.2. GBM Cell-State Analysis

As outlined above, four dynamic cell-states which functionally drive intratumoural heterogeneity within GBM have recently been described [9]. To better understand the contribution of these GBM cell-states within our Q-Cell resource, we firstly analysed 257 unique genes, separating tumours into six metamodules (MES1-, MES2-, NPC1-, NPC2-, OPC-, and AC-like) as per Suva and colleagues [9] encompassing each of the identified four cell-states. We next matched gene expression to the 6172 identified proteins from our MS analysis. Identified proteins corresponded to 38/50 and 29/50 genes from MES1 and MES2, 24/50 and 28/50 genes from NPC1 and NPC2, 26/39 genes from OPC and 30/50 genes from AC-like metamodules respectively (Appendix A). A total of 153 proteins were identified from the corresponding 257 unique cell-state genes outlined by Suva and colleagues.

To identify the contribution of each cell-state in the Q-Cell panel, z-score-scaled protein intensities were assessed for enrichment of the four states using an ssGSEA algorithm [25]. We thus obtained a cell-state score, which was used to predict the predominant cell-state of each model (Figure 2A and Appendix A). Four of the models showed an MES-like state, while two of each model showed an NPC-like and AC-like profile. Four models showed an enrichment of more than one cell-state, termed hybrids. This finding was consistent with Suva and colleagues, highlighting a significant number of GBM cells present in transition states [9]. Principal component analysis showed an enrichment of the predominant cell-state when compared to the rest (Figure 2B and Appendix A). JK2 and MN1 displayed a clear MES-like phenotype matching a total of 39 proteins from the Suva study, with 25 of these proteins showing very high expression (Figure 3 and Appendix A). Similarly, HW1 and FPW1 showed a strong AC-like phenotype with high expression of 15 matched proteins, and RKI1 and WK1 displayed the strongest NPC-like phenotype, while BAH1 showed an OPC-like phenotype (Figure 3). Our analysis identified four of the Q-cell lines, PB1, SHJ1, RN1, and MMK1, to be intermediate hybrids, showing a strong ssGSEA score of more than one cell-state, which is concordant with the results from Suva and colleagues (Figure 3). In addition, we similarly generated cell-state scores for 12 of these cell lines previously analysed by transcriptomics using 252 genes that were identified (Figure 2C and Appendix A) [11]. The proteomics-based and transcriptomics-based cell-state phenotypes for our cell lines were remarkably similar, except for FPW1, which switched from being AC-like to OPC-like.

Our panel of 13 primary GBM cell lines consisted of one pair-matched primary and one recurrent tumour, designated SB2 and SB2b; respectively. SB2 showed an OPC-like state, while SB2b was enriched for the MES-like cell-state with increased expression of marker proteins SERPINE1, GSN, NNMT, SOD2, PLP2, and PRDX6 (Figure 4A,B) with an overall increase of the ssGSEA MES score. We performed a student’s t-test (two-sided, *p*-value cut off >0.05) to identify the significantly regulated subset of proteins between the two cell lines (Appendix A) and then performed enrichment analysis to look for GO molecular functions and KEGG pathways that are among the top 100 most regulated genes in the recurrent SB2b model. The analysis revealed an enrichment of DNA recombination and replication, ECM organisation and cell–substrate junctions, antigen presentation, morphogenesis and organ development, and calcium homeostasis (Figure 3C and Appendix A).

### 3.3. Bioinformatics and Gene-Ontology-Based Protein Analysis

To gain a broader understanding of the differences in biological processes and signalling pathways between Q-Cell models, we performed an analysis of variance (ANOVA) at a 1% permutation-based FDR cut-off (Appendix A). Amongst the top ten regulated proteins were GBM-associated structural and signalling members, including glial fibrillary acidic protein (GFAP), fibronectin (FN1), and Notch1 [29,30,31]. PB1 protein expression correlated the least with the rest of the Q-Cell models (Figure 1) and displays a number of unique morphological characteristics, including high growth and small cell morphology in culture (data not shown). The highest exclusively expressed proteins, identified in PB1, were ubiquitin-conjugating enzyme-2 (UBE2) and SKI. Interestingly, the SKI protein has been associated with high cellular tumours and functions to negatively regulate transforming growth factor beta (TGF-beta) by direct interaction with Smads [32,33].

We next performed a Fisher’s exact test to evaluate the enrichment of KEGG pathways in the ANOVA-positive subset of proteins. Analysis revealed an enrichment of the KEGG term “Pathways in Cancer”, that consists of proteins mediating extracellular matrix (ECM), focal adhesion, and other associated pathways (Appendix A). We extracted all identified member proteins belonging to this pathway and performed unsupervised hierarchical clustering. Results showed a segregation of eight models with higher expression of EMT/-mesenchymal-related proteins BAH1, MMK1, MN1, RN1, SB2b, SB2, WK1, and JK2, termed mesenchymal protein cluster 1 and cluster 2 (Figure 5).

Mesenchymal cluster 1 and cluster 2 consist of structural proteins known to be over-expressed in predominantly mesenchymal-like disease, including laminins, integrins, and collagens, as well as proteins involved in GAP junctions and cell cycle [34]. The remaining Q-Cell panel displayed strong expression of proteins correlated with genes identified from the TCGA GBM neuronal signature and was therefore termed neuronal protein cluster. Models in this cluster showed higher expression of proteins associated with PI3K and mTOR signalling pathways, TCA cycle, NFKB, and MAPK signalling. CDKN2A was found expressed only in MMK1 and RKI1, in concordance with the observation that these were the only lines that did not have a homozygous deletion on the gene level in our characterisation data [10,11]. All individual proteins and their z-scored expression values are provided in Appendix A. We compared the proteome-based expression of proteins shown in Figure 6 to the transcriptomics-based RNA-Seq expression values from our previous study [11]. The Pearson’s coefficient of correlation was low to moderate between mRNA and protein levels for most cell lines and ranged between 0.18 and 0.56, with the exception of PB1, which showed a high correlation of 0.78 (Appendix A). While genes such as *EGFR*, *CDKN2A*, *PRKCA*, and *HSP90B1* showed good correlation between mRNA and protein; others such as *lamininB1*, *NRAS*, and *tenascin* showed a poor correlation. A caveat to our data was that some key GBM-associated genes such as *PTEN*, *MYC* and *MET* were not identified in our proteomics analysis. In addition, we also performed hierarchical clustering of mRNA expression values and found that the Q-Cell panel segregated into two major clusters similar to our proteomics data (Appendix A).

To identify potentially actionable targets that could be either uniquely or ubiquitously expressed in the Q-Cell panel, we matched the OncoKB annotated genes to our dataset. OncoKB is a comprehensive precision oncology database that offers evidence-based drug information on FDA-approved therapies and other investigational agents. We identified 298/642 OncoKB annotated genes in our dataset and extracted the list of proteins identified that corresponded to genes that could be targeted with FDA-approved drugs (Figure 6). Previous therapy studies using the Q-Cell resource have validated our proteomic findings. Johns and colleagues showed that BAH1, which expresses high-level EGFR, effectively responded to gefitinib and panitumumab [35].

## 4. Discussion

Here, we built upon our existing body of work to develop a well-described and characterised open-access collection of relevant GBM models for the neuro-oncology research community. Despite significant effort, there is yet to be meaningful increases in survival for GBM sufferers. Our previous Q-Cell resource characterisation study primarily focused on transcriptome-based mRNA expression analysis [11]. mRNA translation is tightly regulated by numerous processes, and gene expression is not always an accurate predictor of protein abundance [36]. We performed MS-based proteomics analysis of the thirteen Q-Cell models, generating a quantitative and deep proteome abundance atlas representing 6167 human proteins. In our analysis, we observed a number of key GBM-associated genes that were not detected at the protein level and may reflect either a lack of translation or limitations of the MS-based approach used. MS-based proteomics has been used effectively by others to measure the proteome, secretome, and glycoproteins in immortalised GBM cell lines [37,38]. 

Our objective was to further increase the translational significance of the Q-Cell datasets and to provide a platform for researchers to undertake faithful preclinical neuro-oncology studies.

Principal component analysis of Q-Cell replicates grown as 3D neurospheres, showed a high degree of fidelity of protein expression within a given model. All samples in the Q-Cell resource have been generated from IDH1 WT GBM specimens. We found a high degree of common protein expression across the panel, which was expected given the IDH status of the models. Despite this commonality, there were also a number of unique proteins highly expressed in each model.

Elevated expression of common GBM-associated proteins was identified with a trend toward either a broad mesenchymal or neuronal-like signature. An MES-like signature was identified in eight models, while the remaining five models displayed a neuronal-like signature. Given that all models were cultured under the same conditions, this result likely reflects the population of cells captured during the initial cell line generation process. It is worth noting that the majority of Q-Cell models were generated from surgical waste tissue aspirates taken from multiple tumour regions [10]. This was a deliberate attempt to better capture the spectrum of heterogeneous elements within any given tumour, though subclonal elements were likely enriched during the culture generation process.

A major contributing factor to the poor clinical responses observed in GBM patients can be attributed to the significant tumour heterogeneity present within these tumours [8]. Numerous attempts have been made to classify GBM subtypes underlying this heterogeneity [4,5,6,39]. Suva and colleagues have recently extended current thinking and shown that four functional and dynamic GBM cell-states can exist within a single tumour entity [9]. Given this recent finding, we sought to determine the distribution of these cell-states within the Q-Cell resource. Our initial findings highlighted a good representation of each cell-state across the panel, with a number of models expressing multiple cell–state markers at high levels. These models, termed hybrids, likely contain multiple initiating clones arising from different cell-states or might have arisen from a single clone that has two/more major cell-states. Interestingly, of the 257 cell-state genes described in the Suva study, we detected 153 (57%) at the protein level. MS-based proteomics is not exhaustive, and some key proteins may not have been detected in our analysis. Alternatively, some of the cell-state marker genes may not have been translated to protein. Hence, using gene expression data alone may not be the best approach for accurate biological assessment or translational studies. The NPC-like RKI1 model was derived from a patient with the longest survival from the Q-Cell panel. The NPC-like markers in RKI1 were high, resembling a proneural subtype which is associated with a better prognosis. We also analysed a single pair-matched pre- and post-treatment model (SB2/SB2b). Proteomic results generated from this paired model were intriguing and indicated a number of uniquely overexpressed proteins in the recurrent setting. We also noted a shift to a more mesenchymal phenotype post-treatment, a phenomenon previously reported by others [4,12].

To guide potential future translational studies, we identified a number of unique and common targetable proteins using the OncoKB database. These proteins were targeted with either FDA-approved therapies or other investigational agents and could be used to validate precision oncology or cell-state targeting strategies in the future. 

In summary, here we present our detailed proteomics findings of the open-access Q-Cell GBM resource and hope this will accurately guide and inform future studies to improve outcomes for GBM sufferers.

## Figures and Tables

**Figure 1 cells-09-00267-f001:**
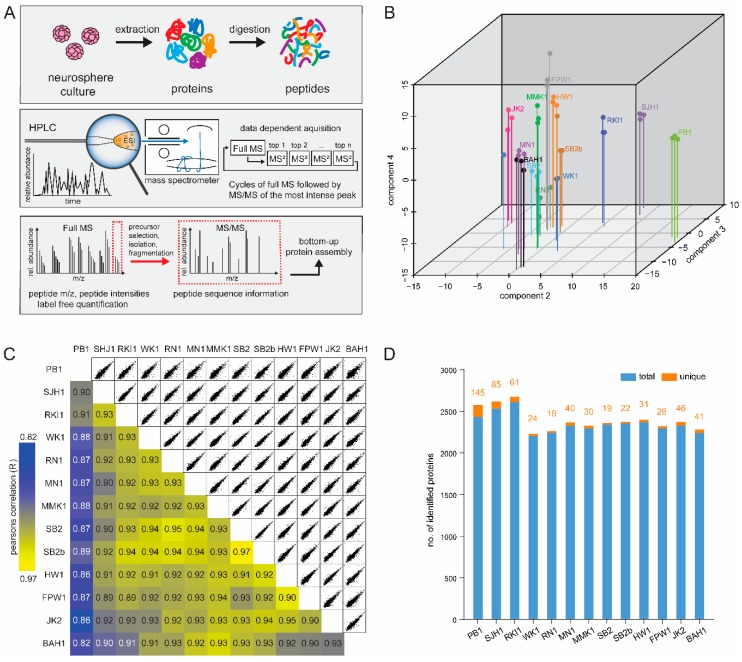
Proteome mapping of 13 Q-cell cell lines. (**A**) Graphical illustration of the workflow for glioblastoma (GBM) cell line proteome analysis. GBM primary cell lines were cultured as neurospheres, lysed in SDS-based buffer, trypsin digested by in-solution digestion in biological triplicate. LC–MS/MS with 4 hour runs and CID fragmentation were performed in the LTQ-Orbitrap Elite. (**B**) 3D-PCA plot. The proteome of thirteen cell lines measured in triplicates segregated into major cell types showing well-correlated triplicates within a cell line. (**C**) The matrix of 78 correlation plots revealed very high correlations between protein intensities in triplicates (Pearson correlation coefficient 0.82–0.97 between cell types). The colour code follows the indicated values of the correlation coefficient. (**D**) A bar chart showing the number of total proteins identified in each of the cell lines with false identification rate (FDR) of 1%.

**Figure 2 cells-09-00267-f002:**
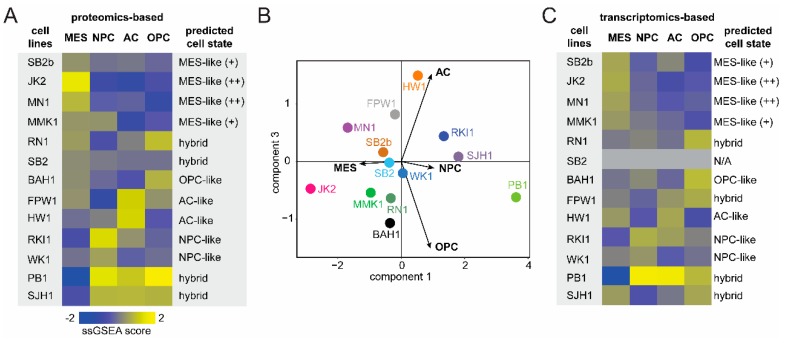
ssGSEA based cell-state scores (**A**) Heatmap of the z-scored ssGSEA scores of the thirteen cell line proteomes with predicted cell-states on the right. (**B**) PCA plot showing the distribution of the cell lines across the four cell-states. (**C**) Heatmap of the z-scored ssGSEA scores of the twelve cell line transcriptomes with predicted cell-states on the right. (Note: SB2 was not analysed by transcriptomics). Abbreviations: MES-like, mesenchymal-like; AC-like, astrocyte-like; OPC-like, oligodendrocyte progenitor cell-like; NPC-like, neural progenitor cell-like.

**Figure 3 cells-09-00267-f003:**
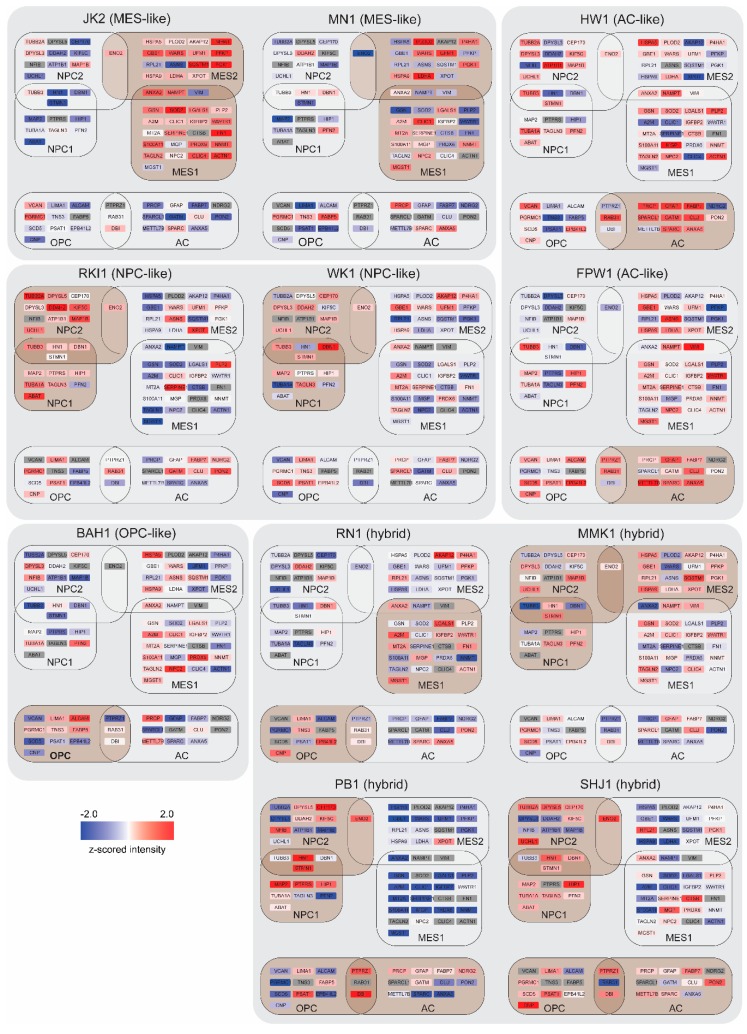
Depiction of cell-states in 11 GBM cell lines. z-scored expression values of genes that contribute to MES1-like, MES2-like, NPC1-like, NPC2-like, and AC-like metamodules (gene list from [9]). Brown boxes indicate predominant cell-state(s) in each cell line. Colour key: red, high expression; blue, low expression; grey, not detected.

**Figure 4 cells-09-00267-f004:**
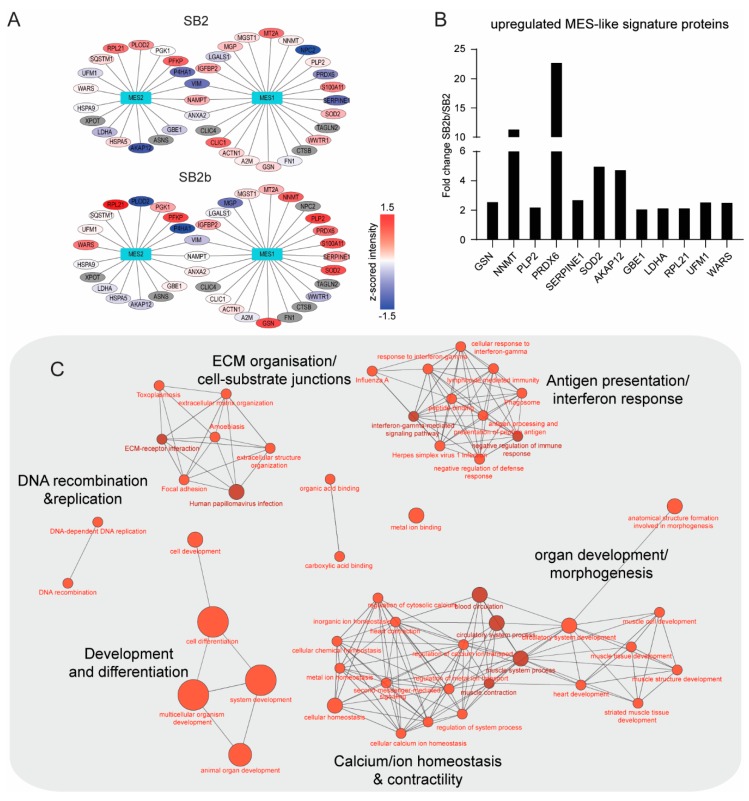
Changing cell-states in a pair-matched recurrent model. (**A**) z-scored expression values of genes that contribute to MES1-like and MES2-like metamodules (gene list from [9]). Colour key: red, high expression; blue, low expression; grey, no expression. (**B**) A bar graph of MES-1-like and MES-2-like genes with >2-fold increase in expression in the recurrent model (SB2b) when compared with the primary model (SB2). (**C**) Kyoto Encyclopedia of Genes and Genomes (KEGG) pathways and gene ontology (GO) molecular function terms enriched in the top 100 most regulated proteins in the recurrent SB2b model when compared to the primary SB2B model. Node size reflects the number of regulated proteins associated with a particular term/pathway. Node colour intensity increases with increasing significance of term enrichment.

**Figure 5 cells-09-00267-f005:**
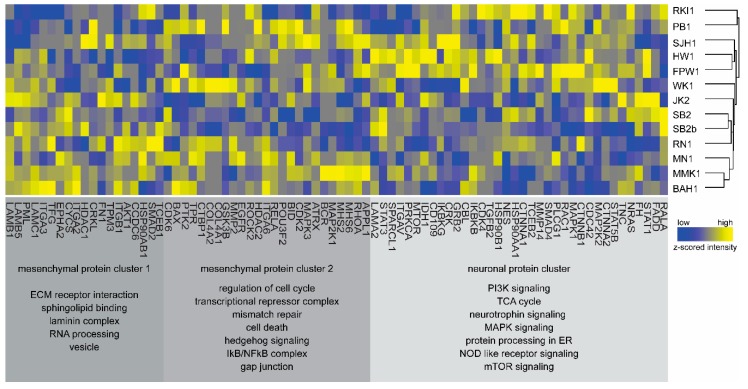
Functional analysis of the expressed proteome. Heatmap of proteins (z-scored intensities) belonging to the KEGG cancer pathway. The proteins segregate into three clusters: mesenchymal cluster 1, mesenchymal cluster 2, and neuronal cluster listing the pathways that proteins represent. Dendrogram depicts the similarity in protein profiles among the 13 cell lines. Colour legend: blue, low expression; and yellow, high expression.

**Figure 6 cells-09-00267-f006:**
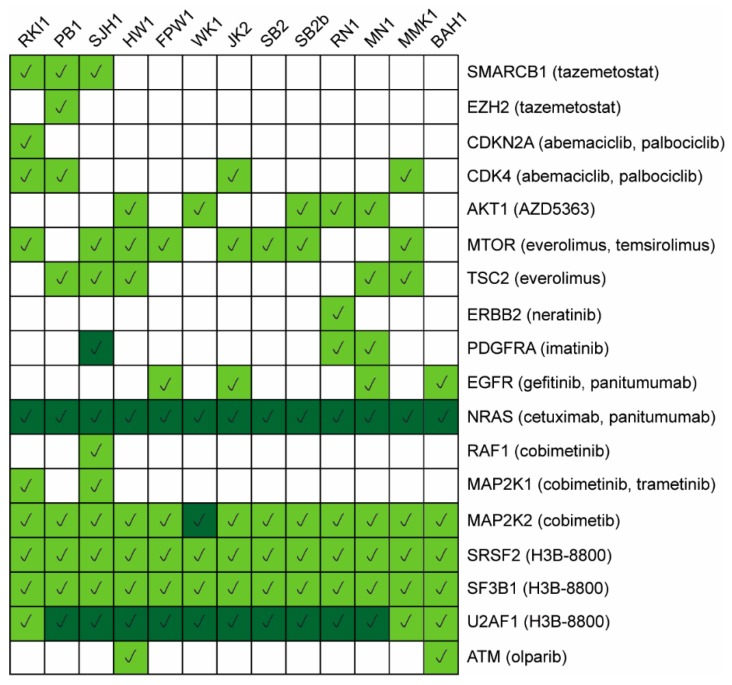
Protein expression matrix of targetable proteins. Expression levels (high, dark green; medium to low, light green; white, not detected) of proteins with potential FDA-approved drugs extracted from the OncoKB knowledgebase.

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
