# Peer review of "Q-Cell Glioblastoma Resource: Proteomics Analysis Reveals Unique Cell-States Are Maintained in 3D Culture"

_cells, 2020, doi:10.3390/cells9020267_

Round 1
Reviewer 1 Report
This paper provides a comprehensive label free proteomic study of a glioblastoma stem cell" resource. The authors report that by proteomics that a number of cell lines display heterogeneity or more recently identified cell states and confirm previous findings of a mesenchymal-like shift following tumor recurrence. Further they sort and provide a list of potential targeted therapies for each cell line. Everything technically appears to have been done correctly. Although many journals, including top proteomics journals, do require that the raw instrument data itself be placed online. Critique: This is really a rather modest contribution to the literature as it clearly states its intention is to be a resource. In that case there are journals that support submission of scientific data. However, they do have some interpretation here. I would expand on a comparison of their transcriptomic data to their newly acquired proteomics data. More specifically, provide some explanation for why the proteomic identifications agree and or disagree with the transcriptionally defined subtypes. This would also be a valuable column to add to Fig. 2A. The authors initially demonstrate a very interesting finding that the in vitro culture appears to maintain the cellular heterogeneity seen in patients yet their final figure is to provide specific drug targets, this appears paradoxical. Minor Comments: Figure 1/Lines 66-70: The pearson correlation between cell lines is r generally not the square, r^2, which did you use? This figure is useful from a technical standpoint but the high correlation is seen in nearly all proteomic studies seen and conducted by this reviewer. This in part has to do with the most of the proteome remains unchanged, generally speaking, and that differentially expressed proteins are outliers. Typically we approx./assume a normal distribution which allows us to use the standard "students t test" to compare two different samples. Figure 3 is useless because it is too small to read. Figure 4. Figures A and B would be better replaced by a labeled volcano plot this would show the statistics (as is currently not shown in B) and clarify the point of the figure. Methods section--please specify minimum number of peptides/unique peptides required for identification. You don't include a statistics section despite having an "ANOVA" column in your supplementary table. In this case what was your multiple test correction approach, if at all. References: References 12 and 13 are not the most relevant references to this sentence in which they are used. Rather I would reuse reference 4, the landmark Phillips 2006 Cancer Cell paper, as frankly little new has been discovered or changed since then
Author Response
We thank all the reviewers for their positive comments regarding our Q-Cell proteomics study.
Response to Reviewer 1 Comments
This paper provides a comprehensive label free proteomic study of a glioblastoma stem cell" resource. The authors report that by proteomics that a number of cell lines display heterogeneity or more recently identified cell states and confirm previous findings of a mesenchymal-like shift following tumor recurrence. Further they sort and provide a list of potential targeted therapies for each cell line. Everything technically appears to have been done correctly. Although many journals, including top proteomics journals, do require that the raw instrument data itself be placed online.
RESPONSE: We can confirm all data has been made publically available online. We could not upload the RAW files to the PRIDE database due to format issues. Therefore, we have uploaded both the RAW files and analysed data to the Q-Cell website. We have amended the methods section accordingly - “All RAW files and protein-based quantification results are available for download from the Q-Cell website at https://www.qimrberghofer.edu.au/q-cell/”
Please expand on a comparison of their transcriptomic data to their newly acquired proteomics data. More specifically, provide some explanation for why the proteomic identifications agree and or disagree with the transcriptionally defined subtypes. This would also be a valuable column to add to Fig. 2A.
RESPONSE: We thank the reviewer for this suggestion. To address this, we compared the cell-state score using our RNASeq versus proteomics data. The cell-state score remained the same in all lines examined except for FPW1. This comparison data has been added as a heat map in Figure 2. We also compared the profiles of the 80 GBM cancer pathway genes analysed in Figure 6. We also added an additional 20 GBM-associated genes showing SNPs in our original Q-Cell paper. The correlation between mRNA and protein was low to moderate across the panel. This data has been added as a supplementary figure (Figure S3). We also provide a transcriptome based heat map for these genes in Figure S4. Text has been modified accordingly.
Minor Comments: Figure 1/Lines 66-70: The pearson correlation between cell lines is r generally not the square, r^2, which did you use? This figure is useful from a technical standpoint but the high correlation is seen in nearly all proteomic studies seen and conducted by this reviewer. This in part has to do with the most of the proteome remains unchanged, generally speaking, and that differentially expressed proteins are outliers. Typically we approx./assume a normal distribution which allows us to use the standard "students t test" to compare two different samples.
RESPONSE: Pearson’s correlation we used in r and we have changed this in Figure 1.
Figure 3 is too small to read.
RESPONSE: We have corrected the resolution of Figure 3 to increase readability.
Figure 4. Figures A and B would be better replaced by a labeled volcano plot this would show the statistics (as is currently not shown in B) and clarify the point of the figure.
RESPONSE: We thank the reviewer for this suggestion. We did generate the volcano plot but found that the data, in our opinion, was not visually enhanced in this format and request that the data be retained in its current format.
Methods section--please specify minimum number of peptides/unique peptides required for identification.
RESPONSE: We have specified the minimum peptide and unique peptides in the methods.
You don't include a statistics section despite having an "ANOVA" column in your supplementary table. In this case what was your multiple test correction approach, if at all?
RESPONSE: We thank the reviewer for pointing this out. We did have a short explanation of the ANOVA analyses but have added more detail. We changed the title of the subsection to read Bioinformatics and statistical analysis.
References: References 12 and 13 are not the most relevant references to this sentence in which they are used. Rather I would reuse reference 4, the landmark Phillips 2006 Cancer Cell paper, as frankly little new has been discovered or changed since then.
RESPONSE: We have removed reference 13 and added Phillips et al to the reference.
Reviewer 2 Report
This manuscript by D’Souza and colleagues report a mass spectrometry characterization of 3D cultured patient-derived glioblastoma cells. This add to the previous RNA expression characterization of the same cell lines that the group has reported and its associated clinical information (References 10, 11 and 14).
Also, the authors performed a comprehensive bioinformatics analysis for cross-validation and comparison of the results that they obtained by the ones recently reported by Neftel and collaborators using single cell RNA-seq, for which they have found a good correlation.
Overall, the data reveals the use of mass spectrometry as a way to characterize cancer-cell subtypes within complex tumor samples, a method of culturing 3D patient-derived tumoroids that reflects the phenotype of cancer cells within the tumors and a way to access and identify novel drug targets for precision oncology. Overall, the results are very well and clearly presented.
Main concerns.
My main criticism is whether would be possible to include some data that validate mass spectrometry data in biopsy sections, for example some of the proteins described in Figure 6, could be a short but interesting list to test this. However, I understand that this might or not be possible due to accessibility to frozen tissue from which the cell lines were obtain as well as availability of commercially available antibodies.
Minor comments.
The resolution of Figure 3 seems to be low in the PDF I received and even lower after printing. This could be improved (increasing slightly the font size of the name of the genes and reducing white space between genes).
Figure S1. The scale of the Y-axis should be the same for the 4 graphs.
Author Response
We thank all the reviewers for their positive comments regarding our Q-Cell proteomics study.
Response to Reviewer 2 Comments
My main criticism is whether it would be possible to include some data that validate mass spectrometry data in biopsy sections, for example some of the proteins described in Figure 6, could be a short but interesting list to test this. However, I understand that this might or not be possible due to accessibility to frozen tissue from which the cell lines were obtain as well as availability of commercially available antibodies.
RESPONSE: We agree that this would be beneficial, however we sincerely apologise that we do not have access to the pair-matched tissue to conduct this analysis.
Minor comments.
The resolution of Figure 3 seems to be low in the PDF I received and even lower after printing. This could be improved (increasing slightly the font size of the name of the genes and reducing white space between genes).
RESPONSE: This has been corrected.
Figure S1. The scale of the Y-axis should be the same for the 4 graphs.
RESPONSE: This has been corrected.
Reviewer 3 Report
This article presents the results of the proteomic analysis of a panel of 13 patient glioblastoma-derived cell lines, recently generated by the authors. Of note, each of these cell lines was previously shown to generate tumors when xenografted into the brain of immunodeficient mice. This cell collection constitutes therefore a relevant resource for exploring glioblastoma physiopathology. The cell proteome profiling was achieved using mass spectrometry (MS), leading to the detection and quantification of the relative abundance of over 2000 human proteins per cell line. Bioinformatics analyses of protein expression profiles based on molecular signatures recently published, further classified the cell lines into differing cellular states resembling mesenchymal (MES), neural progenitor (NPC), oligodendrocyte progenitor (OPC) and astrocyte cells (AC).
Availability of broad and carefully performed protein profiling is rare for such patient-derived cell lines, which are now recognized as powerful tools for relevant modeling of human glioblastoma. This descriptive study provides therefore an interesting resource for the community. Its interest could however be strengthened by providing the following information.
Proteins were extracted from cell lines grown on a laminin substrate, and subsequently cultured under the form of cellular spheres. Such a drastic change in culture conditions is likely to induce an adaptation of the cells to their novel environment. It is therefore important to know at least the cell viability following change of culture protocol. If possible, addition of data allowing to evaluate the degree to which cells adapt their molecular repertoire in response to laminin removal would be a plus. The lists of proteins unique to each cell line should be provided, and commented in the results section. For example, do some of these unique proteins relate to known genomic alterations detected in the cell lines? The authors previously analyzed and published in Scientific Reports the transcriptome profile of the cell lines (Stringer et al, 2019). Since comprehensive proteome profiling cannot be achieved with currently available technologies, knowing the overlap between the transcriptome and proteome profiles of each cell lines would provide valuable insight into the fraction of the cell molecular repertoire accessible with MS/MS proteomic analyses. Likewise, knowing the distribution of the detected proteins according to the levels of their corresponding transcripts would help determining to which extent “…gene expression is not always an accurate predictor of protein abundance.”, as written by the authors at the beginning of their discussion. The authors showed in Figure 5 of their previous paper a Western blot illustrating EGFR expression by all the cell lines analyzed in the present paper. I did not find EGFR in the list of proteins identified in Table S1. Could the authors comment about this discrepant result?Additional comments
The result’s paragraph 3.3 subtitle is misleading. When reading “Functional protein analysis”, I expected to see the results of experiments manipulating the expression or activities of proteins of interest, and the consequences of such manipulations of the cell behaviors. In this paragraph, the authors describe results of bioinformatics and Gene Ontology analyses. The subtitle should be modified according to the type of results described. Lines 282-3, GFAP is described as one of the “structural and signalling members known to promote a stem cell-like phenotype “. To the best of my knowledge, GFAP overexpression has not been shown per se to affect in a positive or negative manner a cell stem-like phenotype in glioblastoma cells. On the opposite, it has been repetitively shown to be overexpressed in Glioblastoma stem cells having lost their stem properties following serum treatment. In the discussion, the authors interpret the fact that only 57% of the 257 genes defined by Neftel and colleagues as cell-state markers on the basis of scRNA-seq analyses, indicates that “not all of the cell-state markers are translated to protein”. Such an interpretation would be possible only if proteomic profiling was exhaustive. This is not the case.Author Response
We thank all the reviewers for their positive comments regarding our Q-Cell proteomics study.
Response to Reviewer 3 Comments
Its interest could be strengthened by providing the following information.
Proteins were extracted from cell lines grown on a laminin substrate, and subsequently cultured under the form of cellular spheres. Such a drastic change in culture conditions is likely to induce an adaptation of the cells to their novel environment. It is therefore important to know at least the cell viability following change of culture protocol. If possible, addition of data allowing to evaluate the degree to which cells adapt their molecular repertoire in response to laminin removal would be a plus.
RESPONSE: We thank the reviewer for this comment. These two conditions are routinely used in the culture of primary brain cancer cells. As part of our previous work we undertook a study to compare these two different conditions and whether they altered the stem cell phenotype or tumourigenicity of the model. Results showed that both conditions are comparable with little variation. We have added this to the manuscript for clarity.
Please see https://www.ncbi.nlm.nih.gov/pmc/articles/PMC4371178/
The lists of proteins unique to each cell line should be provided, and commented in the results section. For example, do some of these unique proteins relate to known genomic alterations detected in the cell lines?
RESPONSE: We have added a column titled “Unique” in Table S1. We did provide one example of the unique identifications matching the genomic alteration data in line 322. “CDKN2A was found expressed only in MMK1 and RKI1, in concordance with the observation that these were the only lines that did not have a homozygous deletion on the gene level in our characterisation data”. We have now expanded on this in the results section to discuss other unique GBM-associated proteins and their relevance.
The authors previously analyzed and published in Scientific Reports the transcriptome profile of the cell lines (Stringer et al, 2019). Since comprehensive proteome profiling cannot be achieved with currently available technologies, knowing the overlap between the transcriptome and proteome profiles of each cell lines would provide valuable insight into the fraction of the cell molecular repertoire accessible with MS/MS proteomic analyses. Likewise, knowing the distribution of the detected proteins according to the levels of their corresponding transcripts would help determining to which extent “…gene expression is not always an accurate predictor of protein abundance.”, as written by the authors at the beginning of their discussion.
RESPONSE: Thank you for this comment. Please see the second response to reviewer 1.
The authors showed in Figure 5 of their previous paper a Western blot illustrating EGFR expression by all the cell lines analyzed in the present paper. I did not find EGFR in the list of proteins identified in Table S1. Could the authors comment about this discrepant result?
RESPONSE: EGFR was in Table S1 but was hidden due to the fact that there was a filter turned on for the Neftel cell states subtype column, this has been corrected and EGFR is now visible. In terms of EGFR expression, we identified high EGFR expression in 6 cell lines (BAH1, FPW1, JK2 and MN1 - >1 replicate) and (PB1 and SB2b - 1 replicate only). In addition, we identified low level EGFR expression in all lines except for HW1, RKI1, SJH1 and SB2. Our proteomics data accurately reflected the western blot shown in Stringer et al 2019.
Additional comments
The result’s paragraph 3.3 subtitle is misleading. When reading “Functional protein analysis”, I expected to see the results of experiments manipulating the expression or activities of proteins of interest, and the consequences of such manipulations of the cell behaviors. In this paragraph, the authors describe results of bioinformatics and Gene Ontology analyses. The subtitle should be modified according to the type of results described.
RESPONSE: We have modified this section to read ‘Bioinformatics and Gene Ontology based protein analysis’
Lines 282-3, GFAP is described as one of the “structural and signalling members known to promote a stem cell-like phenotype “. To the best of my knowledge, GFAP overexpression has not been shown per se to affect in a positive or negative manner a cell stem-like phenotype in glioblastoma cells. On the opposite, it has been repetitively shown to be overexpressed in Glioblastoma stem cells having lost their stem properties following serum treatment.
RESPONSE: We apologise for this mistake, we agree that GFAP is more commonly associated with GBM cell-differentiation. The reference to a stem cell phenotype has been removed from the text.
In the discussion, the authors interpret the fact that only 57% of the 257 genes defined by Neftel and colleagues as cell-state markers on the basis of scRNA-seq analyses, indicates that “not all of the cell-state markers are translated to protein”. Such an interpretation would be possible only if proteomic profiling was exhaustive. This is not the case.
RESPONSE: We agree, we have re-written this section to state that proteomics analysis is not exhaustive.